# Planting Arrangement and Effects of Planting Density on Tropical Fruit Crops—A Review

Mohammad Amdadul Haque [1,2] and Siti Zaharah Sakimin [1,*]

1   Department of Crop Science, Faculty of Agriculture, Universiti Putra Malaysia, Serdang 43400, Malaysia;
    amdad80@gmail.com
2   Horticulture Research Centre, Bangladesh Agricultural Research Institute, Gazipur 1701, Bangladesh
*   Correspondence: szaharah@upm.edu.my; Tel.: +60-127847290

**Abstract:** With the growing importance of fruits as valuable food resources, attention has been paid in recent years to enhance productivity and quality. Increasing productivity per unit area through agronomic management is one of the important strategies to increase the production of fruit crops. Keeping this view, this review was undertaken to improve understanding of the planting arrangement and the effect of planting density on productivity and quality of fruit crops. This review will thus provide updated and in-depth information about the influence of planting density on yield and fruit quality, which will also be useful for the growers, consumers, exporters, and processing industry. In fruit crops, the effect of plant density and planting arrangement is well documented. From the review, it is understood that yield per unit area is the product of fruit yield plant$^{-1}$ and the number of plants per unit area. Fruit size increases with a decreasing planting density, while total yield increases with an increasing density at a certain level. However, fruit quality decreases with an increasing planting density.

**Keywords:** planting arrangement; planting density; fruit crops





## 1. Introduction

Fruits are an excellent source of essential vitamins and minerals, and they are high in fiber. Fruits also provide a wide range of health-boosting antioxidants, including flavonoids [1]. The consumption of fruits can reduce a person's risk of developing heart disease, cancer, inflammation, and diabetes. By 2050, the world's population will be around 9.1 billion, a 34 percent increase over the current population [2]. Ultimately, the demand for food is predicted to rise at a similar rate [3]. Furthermore, due to a shortage of resources combined with urbanization, the degradation of arable land, and shortage of water is a key bottleneck in fruit production. To combat this situation, the use of fertilizers and pesticides, improved agronomic management, and high-yielding, insect-, and disease-resistant crop varieties have been in great mandate with the farmers/growers for the past five decades. Agronomic management has played a crucial role in improving the productivity of crops and fruits in particular. The World Health Organization (WHO), and the Food and Agriculture Organization (FAO) recommend a minimum daily consumption of 400 g of vegetables and fruits a day, excluding starchy tubers. However, the statistics of Food and Agriculture Organization and World Health Organization state that the worldwide per capita consumption of fruits and vegetables is estimated to be less than 20 to 50 percent of the minimum daily recommended intake [4]. The low consumption is mainly due to a very small fraction of the total cropped area under fruits, including the unavailability of fruits due to lower yield and a poor diet of junk foods that are high in calories and low in nutrients. The gap between recommended daily allowance (RDA) and consumption can be minimized by increasing area and unit area production. Various crop management interventions can be employed for improving the yield and quality of different crops [5]. These include crop

husbandry factors (e.g., seeding rate, spacing/plant density, time of sowing, fertilizer application, other cultural practices); stress factors (abiotic: drought, salinity, temperature, day length; biotic: diseases and pests); external chemical application; intercropping; genotypic factors within the crop. Among the interventions, planting arrangement and planting distance is an important agricultural activity to capture environmentally efficient natural resources and plays a major function in regulating competition between neighbor plant canopies [6].

Plant density refers to the number of individual plants per unit of ground area. In an ideal plant population, the capability of the plant canopy to collect environmental resources, including radiation energy, water, and inorganic nutrients, can be improved [7]. A dense plant population increases the competition between plants for resources, which results in limited resources being depleted [8]. Additionally, improved canopy construction can lead to an optimal leaf area index that can boost photosynthetic ability through efficient solar radiation interception [9]. High densities of plants encourage better productivity; lower densities, in general, allow for the harvesting of more large fruits [10], resulting in higher pricing on the fresh fruit market. Higher planting density is used for higher yields without increasing production costs [11]. The selection of the appropriate density is critical for optimizing the yield potential. The optimal plant density should be kept for the maximum possible yields of good quality fruit. Increased plant population density improves the interception of solar radiation and water use efficiency [12], which helps to boost canopy photosynthetic capacity and biomass production [13], thus increasing yield and water productivity [10]. As a result of the optimal planting density, crop yield and quality are improved, while the need for fertilizer and labour is reduced [14]. However, when planting density exceeds a certain threshold, the yield tends to be reduced [15] due to an inadequate supply of carbon and nitrogen, because of intense interplant competition for sunlight, soil nutrients, and water. In modern cropping systems, high plant density in planting is one of the most important agronomic approaches [16]. Maximum yield occurs when the spatial plant density allows for the maximum leaf area index to be given by the rapid growth of the leaf canopy, enabling maximum solar radiation interception as early as possible in the growing season [17]. Plant population density has been a vital variable in yield enhancement studies [18]. As plant density increases, the distribution of resources on the plant scale (soil water, fertility of the soil, light interception) decreases, increasing competition between plants and, in turn, reducing the potential of individual plant yield [19]. The probability of negative effects increases when the density reaches the optimal level for a given area [20]. One of the best management practices to increase yield is the selection of sufficient planting density to maximize light interception, as crop productivity is highly influenced by the total solar radiation intercepted by the crop. Generally, with rising planting density, and solar radiation obtained by the crop increases, the higher planting density is also found to have higher total yields [10]. High plant density associated with shading may lead to disease infestation, the shedding of fruit, reduced fruit size, delayed maturity, and decreased individual plant growth and light interception [21].

Crop spatial arrangement is another agronomic component that might affect the yield [22] and crop competition against weeds [23]. This pattern can also affect plant growth, and development [24]. Uniform planting patterns reduce mutual shade and hasten canopy closure in increasing the leaf area index (LAI). All of these factors contribute to the canopy's improved radiation interception [25], which boosts crop growth and production [26]. The equal spacing of crops, on the other hand, may restrict light penetration or change the illumination quality under the crop canopy. Some weed seeds may not germinate, weed seedlings may not flourish, and weed seed production may be reduced [26]. When there is a uniform distribution, intraspecific competition within a crop is prolonged, but interspecific competition with weeds begins sooner. This permits the crop population to shade out the weeds and prevent them from re-emerging in large numbers [27]. Planting arrangement, or planting pattern, might be an important determining factor for individual plant performance [28]. The planting arrangement of the crop is the most important

management aspect that determines the crop structure and can alter crop yield–density relationships [29]. Osaigbove et al. [30] mentioned that cropping could be intensified with proper plant arrangement, as this will not only enhance the productivity of the land but also prevents weeds growing in the field. The use of improved and appropriate spatial planting arrangements would facilitate lessening inter-specific competition and maintain sufficient resource capture in the cropping system [31].

In recent years, many types of trial have been conducted to increase the yield and quality of fruit crops, such as the application of organic and inorganic fertilizer, using higher planting density with different planting arrangements, pruning of fruit trees, and fruit thinning. Of these, planting density is considered to be the most suitable technique that influences the yield and quality of fruits. The present review will therefore discuss the effect of planting density on fruit yield and quality, as well as identify the research gaps for future consideration.

## 2. Planting Arrangement of Fruit Crops

Any layout of planting should aim at providing a maximum plant population ha$^{-1}$, suitable space for proper growth of the tree plants, and confirming convenience in orchard management [32]. The layout scheme of the planting pattern can be classified into two groups, viz., (a) vertical row and (b) alternate row. In the former pattern (square and rectangular systems), the trees planted in a row are exactly perpendicular to those planted in neighbouring rows. The trees in the neighbouring rows in the latter pattern (hexagonal, quincunx, and triangular) are not perfectly vertical; instead, the trees in the even rows are halfway between those in the odd rows. The spatial arrangement was highly significant, concerning the plant height, stem girth, and canopy width of a plant [33]. On the other hand, Singh et al. [34] revealed that various planting systems exhibited significant effects on the growth, yield, and quality parameters of fruit. In general, yield components and productivity are affected by the spatial arrangement of the crop, which is determined by a combination of row spacing and plant spacing within the row [35]. The first step in establishing a successful fruit orchard is to create a proper layout plan [36]. Planting arrangements should be thoroughly planned because they determine the cultivation, management practices, plant health, yield, and quality of fruit [37]. The distribution pattern and intensity of light interception are also influenced by how trees are arranged. The arrangement of planting is one of the most effective ways to ensure that land is used efficiently and profitably [38]. Inappropriate spacing has been assigned as the main reason for a lower productivity of fruit crops [39]. Therefore, the selection of optimum plant spacing and a planting system for the effective utilization of land and solar energy (PAR) is of prime significance to obtain good quality fruits and yield [40]. The planting scheme selection is influenced by soil and environmental conditions, as well as the topography of the orchard location, the plant species to be planted, and the orchard management procedure [41]. Planting systems are also known as "layout systems", because no single system is appropriate for fruit plant planting in all situations. For planting fruit crops, the following planting systems are often employed:

### 2.1. Vertical Row Planting Pattern

This pattern of planting is divided into two methods: square and rectangular. The square pattern of planting is the most common method, and it's simple to set up in the field. Plant–plant and row–row distances are the same in this method (Figure 1). The plants are at right angles to each other, forming a square with each unit of four plants. After the orchard is planted, this system facilitates interculture in two directions. The main flaw in the system is that the space in the centre of the square is left unutilized [42].

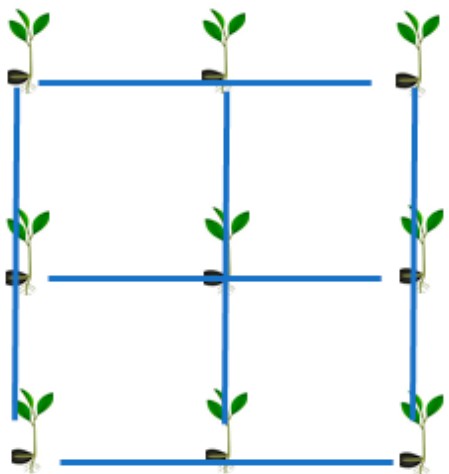

**Figure 1.** Square pattern of planting.

As opposed to squares, the plot is divided into rectangles, and trees are planted in straight rows running at right angles to each corner of the rectangle (Figure 2). Similar to the square system of planting, this system also allows intercropping in two directions. Because the distance between any two rows is higher than the distance between any two trees in a row, there is no equal distribution of space per tree. The main disadvantage of this system is that it results in a greater loss of income when intercropping isn't used [41]. Weed growth is higher than in the square arrangement [43]. The following formula can be used to calculate the number of plants accommodated:

$$\text{Total number of accommodated plants} = \frac{\text{Total area of the land}}{\text{Distance between plants} \ \times \ \text{Distance between rows}}$$

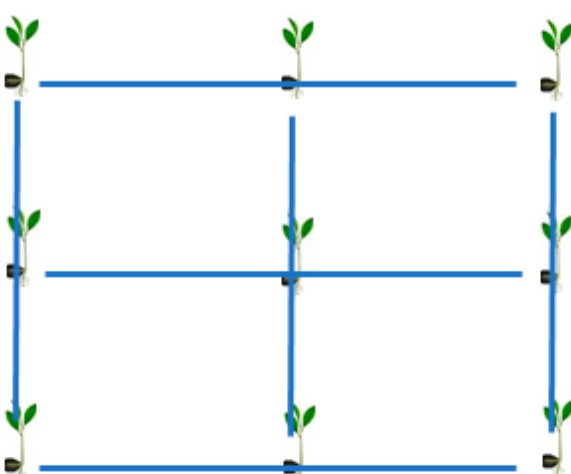

**Figure 2.** Rectangular pattern of planting.

### 2.2. Alternate Row Planting Pattern

### 2.2.1. Hexagonal Pattern of Planting

Within this system, trees are planted in equilateral triangle corners. Six trees are arranged in a hexagon, with another tree in the centre of the hexagon (Figure 3). Though it is a bit difficult to implement, it allows for 15% more plants. With this system, it is possible to cultivate land in three directions between the plant rows. In general, this system is used when the land is expensive, fertile, and irrigation water is abundant. If the distance

between rows and plants remains the same, the hexagonal system could accommodate 15% more plants than the square system [44].

$$\text{Total number of plants required} = \frac{\text{Total area of the land}}{\text{The area occupied by single plant}}$$

$$\text{The area occupies by individual plant} = \frac{3}{4} \times A \times A \times 2$$

where, A = Triangle's side length or plant spacing

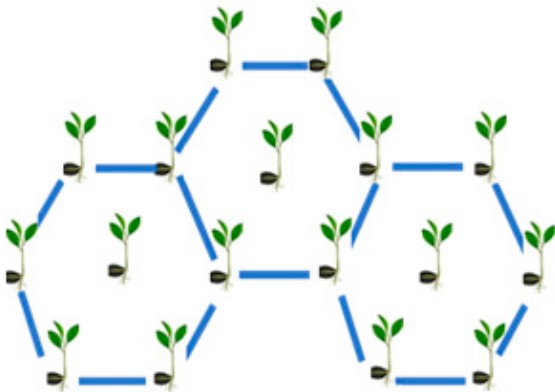

**Figure 3.** Hexagonal pattern of planting.

2.2.2. Triangular Pattern of Planting

The trees are laid out in a square pattern. The trees in even-numbered rows are situated halfway between those in odd-numbered rows, rather than being opposite to each other. The perpendicular distance between any two consecutive rows is equal to the distance between any two trees in a row (Figure 4). This system allows more open space for plants and intercrops, which is beneficial to both. The six plants are planted at each corner of the hexagon, with a seventh plant placed in the middle. There is a slight (11%) reduction in plant accommodation when compared to the square system [44].

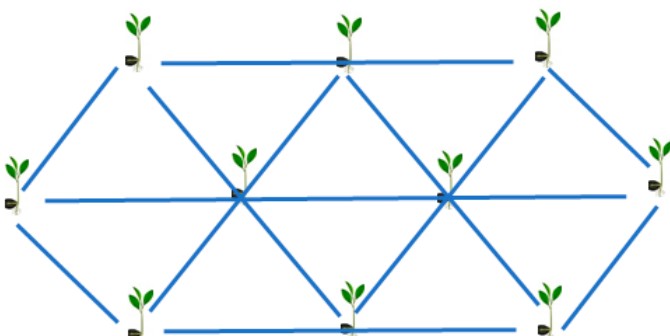

**Figure 4.** Triangular pattern of planting.

2.2.3. Diagonal or Quincunx Pattern of Planting

There are no differences between this system and the square system, except that each square has an additional tree planted in the middle (Figure 5) [45]. This system doubles the number of plants ha$^{-1}$ when compared to the square system. Fruit trees such as papaya, phalsa, kinnow, guava, plum, peach, etc., are planted in the orchard as fillers to provide an additional income to the grower. When the main orchard trees begin bearing fruit, the filler trees are uprooted. A primary fruit crop refers to the main crop, while a secondary or supplementary fruit crop refers to the filler crop. Quincunx planting has the disadvantages of extreme competition for water, nutrients, sunlight, and air between primary and filler

fruit crops, overlapping branches and overcrowding of trees, as well as the difficulty of mechanization of some cultural practices. More plants can be accommodated for in this arrangement, rather than in the main square or rectangular pattern. The following formula can be used to calculate the number of plants required to occupy the available land:

$$\text{Total number of required plants} = \frac{\text{Total area of the land}}{\text{Distance between plants} \ \times \ \text{Distance between rows}} \times 2$$

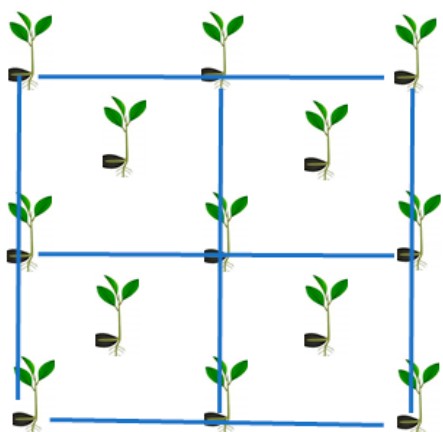

**Figure 5.** Quincunx pattern of planting.

### 2.2.4. Contour Pattern of Planting

Hilly areas with steep slopes typically use this system [45], but the system is very similar to the square/rectangular system in terms of its basic design. Planting the trees in lines that follow the contour of the soil with a slight slope may be the best option in such situations (Figure 6). As a result, irrigation and cultivation are limited to the slope of the land, which reduces soil erosion. A baseline is established at the lowest level and then trees are marked from the base to the top, as in a square/rectangular system. The use of bench terraces is necessary when the slope is greater than 10%. Between rows of fruit trees, shrubs and grass can be planted to help limit water flow, weed growth, and soil erosion [46]. The following equation is used to calculate the number of plants to be accommodated.

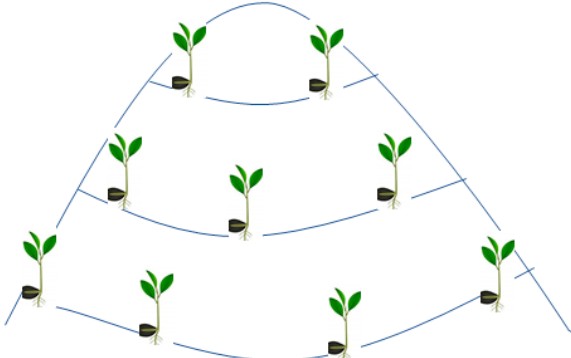

**Figure 6.** Contour pattern of planting.

$$\text{Plant population} = \frac{\text{Number of hedges} \ \times \ \text{Unit area}}{S \ (a + b)}$$

where,
　　S = Spacing between hedges
　　a = Space between hedge

b = Vertical space between rows

The number of plants accommodate in different planting arrangements at diverse planting densities of fruit cops is presented in Table 1.

**Table 1.** Planting distance of different fruit crops and number of plants accommodate in different planting arrangements.

| Crop | Planting Distance (m) | No. of Trees per Hectare | | |
|---|---|---|---|---|
| | | Square Planting | Hexagonal Planting | Triangular Planting |
| Mango | 10 × 10 | 100 | 115 | 89 |
| Sapota | 8 × 8 | 156 | 118 | 139 |
| Clove | 6 × 6 | 277 | 320 | 248 |
| Acid lime | 5 × 5 | 400 | 461 | 357 |
| Coconut | 7.5 × 7.5 | 177 | 205 | 159 |

Source: [46].

## 3. The Benefit of Planting Density

Nutrients are required by plants. The sun, the soil, and any amendments or fertilizers applied to the soil during the season provide these nutrients. Plants that compete with their neighbours for soil nutrients and sunshine will not be as healthy as those that have all of the resources they require. Their roots will also have to compete for space, nutrients, and water. Too few plants can also be a problem. The shade from properly positioned plants can crowd out weeds as they grow and keep the soil moist, creating a favourable environmental condition that only works well if plants are spaced apart suitably [47]. Adequate space between plants will reduce competition for light, will conserve water, and will provide more soil nutrition to each plant [48]. Plant spacing is significant because it reduces disease risk in two ways: contagious disease and immune system improvement [48]. Plant diseases can easily spread from one plant to another when they grow too close together. Plants that grow too close together are less healthy than those that have enough separation [49]. Overcrowding also lowers air circulation and increases the likelihood of plants being sick. For many pathosystems, microclimatic changes resulting from higher plant densities usually create conditions that are more conducive to disease development [50]. Increased plant density is likely to increase the duration of wetness on host tissues.

It is critical to keep weeds under control in order to keep your plants healthy. Planting too close together makes weed control more difficult, which leads to more weeds [51]. When plants are spaced correctly, harvesting becomes much easier [52]. If plants are planted far apart, they will not have to compete at all for resources. They may produce higher yields per plant than more crowded plants, but if they are spaced too far apart, the yield for the entire field of plants can still be low [53]. Appropriate spacing results in higher quantities of tastier, healthier crops.

## 4. Factors to Be Considered for Planting Density

The optimal spacing for crops depends on several factors. The first factors to consider are environmental factors. The right establishment method, plant spacing, and planting configuration of crops depend on environmental conditions [54]. In a shady site, leafy plants may require more space than they do when grown in full sun. It is especially important to avoid overcrowding in shady conditions, where disease may be more likely to take hold. However, plants grown in full sun conditions will typically need more water, and so spacing may need to be increased to take into account the amount of water each plant will require and how much can be provided by the soil or growing medium. The planting density of a crop varies on the rainfall pattern of that area. Low rainfall areas should have wider tree spacing than high rainfall ones [55]. The planting density of the crop also depends on the harvesting stage of the crop. If the plants remain in the growing area to

maturity, they will require a lot more space than crops that are to be harvested at an earlier stage in their development [56]. Wider spacing should be used for bigger or spreading canopies, while narrower spacing can be used for smaller or erect type canopy [46]. For example, mangoes are planted at a distance of $10 \times 10$ m, guavas at a distance of $5 \times 5$ m, and papayas are planted at a $2 \times 2$ m distance. Tall crop varieties require wider spacing, while short varieties require closer spacing [56].

As root and shoot growth is restricted in heavy soils, less spacing should be used. Plant density varies considerably depending on the condition of the growing area and soil fertility status. Fertile soil can support a high plant population [57]. Therefore, closer spacing is possible. Trees trained on the head system require closer spacing than the other types of training systems [48]. On the other hand, the space between the rows should allow free passage for crop management practices. The spacing between rows should be greater for the operation of heavy machinery, while a narrow space is supposed to be used for small machinery [58]. Planting density also depends on the mechanization level of some cultural practices. High planting density can improve the adaptability of mechanized harvests by regulating plant structure [59]. The use of dwarf rootstock is another factor to be considered for planting density as most of the fruit crops are grown with the use of rootstocks. Robinson [60] stated that dwarfing rootstocks have been the key to the dramatic changes in tree size, spacing, early production, and quality of fruit crops. Laužikė et al. [61] evaluated the effect of rootstock and high-density orchards on cv. Auksis fruit quality and reported that super-dwarfing with P22 rootstock planted at $3.0 \times 0.75$ m distance accumulated a significantly higher (up to 45%) content of organic acid and up to 33–44% lower DPPH free-radical-scavenging capacity when compared to P60 dwarfing rootstock planted at a $3.0 \times 1.0$ m distance. Hence, the dwarf rootstock is recommended for close plantations.

## 5. Impact of Plant Density on the Use of Resources

Plants have extreme plasticity in terms of their size and shape that react remarkably to environmental conditions. The existence of competitive neighbours is one of the powers of such external forces and can contribute to a reduction in the size of a plant. Water, nutrients, light, oxygen, carbon dioxide, and pollinating agents during the reproductive process are the factors for which competition takes place between plants. The most common deficiencies are water, nutrients, and light. Competition depends on resource availability and starts when the immediate supply of a single essential factor is smaller than the plants' combined demand [62].

For a single species, the rise in density increases the return per unit area, and intraspecific competition increases as more plants compete for the same common resource constraints. At pure stands, the increase in competition is illustrated by a decrease in the output of individuals, e.g., average individual plant biomass and/or yield [63]. According to Assefa et al. [64], both too small and too big distances affect crop production through competition and shading. When spacing is too large, the yield can be decreased, as growth factors are not being used efficiently. Normally, the increase in density often proportionally increases yield but decreases after certain levels [65]. To achieve maximum performance, it is important to set the optimum density per unit area [6]. A higher density is required to allow for the efficient use of all growth factors in appropriate conditions of soil moisture and nutrients. Each growth factor for which the plant competes cannot support a crop beyond the optimum plant density level per unit area. The plant density level should be such that the maximum solar radiation is used. When planted at a greater distance, the maximum yield capacity of a single plant is completely exploited [39]. Equidistant inter- and intra-row spacing provides soil coverage that is earlier than other distances, thereby more sunlight is intercepted by the foliage. Theoretically, a square system of planting can use energy more effectively than a rectangle [66]. Closely spaced and rapidly growing crops can intercept more light and give higher yields in contrast to wider-spaced crops. The optimum plant density is necessary for the correct light interception in different growth

stages. The greater level of interception increases photosynthesis and decreases soil water evaporation [66].

The planting arrangement changes the temporal and spatial pattern in which the limiting resource is intercepted, especially for crops in drylands, in which soil water is rarely sufficient during the growing season. In such circumstances, the spacing between and within the rows is usually a concern [67]. Closely spaced plants have an increased elongation of their roots and thus continue to extract moisture available between the rows for grain production later in the season [68]. The density of the plant must be adjusted to the moisture available, either within the rows or between the rows.

The shadow of the soil surface by a crop canopy affects soil evaporation directly [69]. Suzuki et al. [70] found a higher water use efficiency when the crop canopy was larger. Increasing plant density is one way to achieve a dense canopy that will therefore make the water more efficient [71,72].

## 6. Planting Density on the Growth and Development of Fruit Crops

The plant growth can be visualized in terms of an increase in length, plant height, or stem diameter, an increase in fresh and dry biomass weight, an increase in leaf area and leaf weight, and so on [73]. The density of the plantation has a major influence on plant growth and development. The magnitude of the effect usually depends on whether plants compete with one another and on the pattern of growth and morphological characteristics of the competitor [74]. A high density causes certain changes in plant growth, e.g., plant height increases, leaf thickness decreases, leaf orientation changes, and the leaves become upright, thin, and spaced at longer intervals in a vertical order to intercept more sunlight [75]. Onat et al. [76] reported that as the plant density was increased per unit area, plant height increased substantially. Hung et al. [77] reported on a pineapple variety smooth cayenne in which plant height and leaf number increased with the increasing planting density. Malézieux et al. [78] reported a significant decrease in the specific leaf weight of pineapple when planting density was greater than 6 plants m$^{-2}$. Leaf size and shape influence the amount of light interception, all of which differ substantially with the plant density and inter- and intra-row spacing [79]. Increased planting density increases shading and delays the maturation of both leaves and seeds. Research conducted in the Philippines demonstrated that pineapple plants can generate more specific leaf weight and large amounts of biomass in abundant space [80]. However, Mabapa et al. [81] observed that the increase in planting density led to an increase in plant biomass per unit area. Moreover, above-ground biomass is directly related to plant density, where light use efficiency was increased with the increasing plant density [82]. A proper increase in plant density can provide an adequate population and can maintain relatively high photosynthetic ability and delayed leaf senescence during late growth and development, in order to maintain dry matter supply [83]. The planting of pineapple at different planting densities revealed significant differences in its leaf size index. A decrease in leaf width, D leaf length, and leaf size index of the Kew pineapple with increasing planting density was reported by Laishram et al. [84] in India. Authors in [85] have reported, on the other hand, that the stem diameter of pineapple increases with a decreasing plant spacing/increasing planting density.

Leaf area regulates the light interception capacity of a plant and is often used as a surrogate for its growth [86], which is very important parameter in determining plant productivity. Streck et al. [87] reported that green leaf area plant$^{-1}$ was higher in treatments with larger spacing. The highest leaf area plant$^{-1}$ from the treatment of wider spacing was due to the wider and higher number of the leaf. The index of leaf area (LAI) represents crop leafiness and is an important crop growth parameter [88]. All agricultural practices (planting spacing and population densities) lead to alterations of crop canopy and modification of leaf area index [88]. In general, highly populated plants tend to close the soil faster than smaller communities, and therefore, in densely populated plants, the optimum leaf area index is typically achieved more quickly when compared to sparsely populated plants. Planting density influences the leaf area index of crop plants [89]. Because of the

larger density, the LAI per pit gradually increased. Although the size and number of leaves on individual plants reduced in high-density planting, the total number of plants and leaves per unit area increased dramatically, resulting in a considerably higher LAI [90]. The effect of high-density banana planting on leaf area and LAI was similarly documented by Ladaniya et al. [91]. Acid lime grown under a high density had a dense and vigorous plant canopy and recorded a maximum leaf area index when compared to a lower density [92] which was supported by the findings of Santos et al. [93] on papaya. The effect of planting density on the growth, yield, and quality of some fruit crops is shown in Table 2.

**Table 2.** Effect of planting density/spacing on growth, yield, and quality of different fruit crops.

| Sl. | Name of Fruit Crops | Planting Spacing/Density | Key Findings | Reference |
|---|---|---|---|---|
| 1 | Dragon fruit (*Hylocereus polyrhizus*) | 1815 plants ha$^{-1}$<br>1556 plants ha$^{-1}$<br>1361 plants ha$^{-1}$ | Cumulative data over four years showed that the highest density at 1815 plants ha$^{-1}$ produced 48.7 t ha$^{-1}$, followed by 41.8 t ha$^{-1}$ and 38.2 t ha$^{-1}$ with the planting density of 1556 plants ha$^{-1}$ and 1361 plants ha$^{-1}$, respectively. | [94] |
| 2 | Sweet orange (*Citrus sinensis*) | 358 trees ha$^{-1}$<br><br>955 trees ha$^{-1}$ | The findings over seven years showed that higher density (955 trees ha$^{-1}$) resulted in greater canopy volume on an area basis, which explained the 86% to 300% increase in cumulative fruit yield ha$^{-1}$ that resulted from lower (358 trees ha$^{-1}$) to higher tree density (955 trees ha$^{-1}$). Soluble solids content (SSC), titratable acidity (TA), and SSC: TA ratio was the highest under the lowest (358 trees ha$^{-1}$) density in 2016–2017, with no treatment effects on quality parameters in other years. | [95] |
| | | 3.30 m × 6.60 m<br>3.30 m × 3.30 m<br>6.60 m × 6.60 m | Plant height was maximum at 3.30 m × 3.30 m plant spacing but the number of leaves was higher at 3.30 m × 6.60 m. Maximum fruit yield plant$^{-1}$ (3.63 kg) was observed from the spacing of 3.30 m × 6.60 m while minimum yield (2.11 kg plant$^{-1}$) was recorded on a closer plantation (3.30 m × 3.30 m). | [96] |
| | | 4.0 m × 4.0 m<br><br>5.0 m × 5.0 m<br>6.0 m × 6.0 m | Plants at 6.00 m × 6.00 m spacing exhibited better tree growth and improved the size and weight of fruit than at 5.00 m × 5.00 m and 4.00 m × 4.00 m spacings. Plant spacing did not affect yield plant$^{-1}$ but closer spacing at 4.00 m × 4.00 m produced more fruit yield ha$^{-1}$ than others. The planting distance of sweet orange trees did not influence the quality of the fruits. However, the highest TSS (18.21% Brix) was recorded from wider spacing (5.00 m × 5.00 m) | [97] |
| | | 2.40 m × 4.50 m<br><br><br>4.50 m × 6.00 m | Trees at the 2.40 in-row spacing reduced trunk growth. Fruit size was influenced by spacing. Fruits were smaller on trees at the closer in row and between row spacings. Yield increased with the increasing tree density during the 4th and 5th seasons. A higher yield of oranges was recorded at a closer intra-row of 2.40 m (99 box ha$^{-1}$) and inter row of 4.50 m (83 box ha$^{-1}$) compared to the lower yield at a higher intra-row spacing of 4.50 m and inter-row spacing of 6.00 m. Fruit quality (TSS, % acidity, % juice content) was not influenced either by intra or inter-row spacing. | [98] |
| 3 | Mandarin (*Citrus reticulata*) | 277 plants ha$^{-1}$<br>555 plants ha$^{-1}$<br>625 plants ha$^{-1}$<br>1250 plants ha$^{-1}$<br>2500 plants ha$^{-1}$ | The plants were tallest at 2500 plants ha$^{-1}$ with the lowest leaf area index. Fruit length and breadth were significantly higher at lower density (277 plants ha$^{-1}$). Fruit yield plant$^{-1}$ was highest at 277 plants ha$^{-1}$ but fruit yield ha$^{-1}$ under 12500 plants ha$^{-1}$ was 26, 7.1, and 4.6 times more compared to conventional planting during the 1st, 2nd, and 3rd year, respectively. Fruit TSS (9.57% Brix) and juice acidity (0.81%) were higher at lower densities of 555 and 625 plants ha$^{-1}$, respectively. | [99] |
| | | 3.30 m × 6.60 m<br>3.30 m × 3.30 m<br>6.60 m × 6.60 m | The maximum (26.98) number of fruits plant$^{-1}$, fruit diameter (6.55 cm), average fruit weight (141.11 g) with maximum fruit juice percentage (51.48%), and reducing sugars (2.30%) were observed in 3.30 m × 6.60 m spacing. Highly dense plantations at 3.30 m × 3.30 m resulted in the highest ascorbic acid (6.85 mg 100 gm$^{-1}$), total sugars (8.50%), and non-reducing sugars (7.21%). | [100] |
| | | 6.00 m × 6.00 m<br>6.00 m × 5.00 m<br>6.00 m × 3.00 m | Vegetative growth was higher at greater spacing (6.00 m × 6.00 m). The maximum number of fruit (276) and yield plant$^{-1}$ (46.65 kg) was found in low density (6.00 m × 6.00 m), while yield ha$^{-1}$ was the highest (220.99 t ha$^{-1}$) at the closest spacing (6.00 × 3.00 m). | [39] |
| | | 3.30 m × 6.60 m<br>3.30 m × 3.30 m<br>6.60 m × 6.60 m | Maximum plant height 3.49 m) and the number of leaves (80) were observed in closer (3.30 m × 3.30 m) spacing. Maximum yield was found in 3.30 m × 6.60 m spacing. Higher TSS was observed in wider spacing, while juice percentage was observed in closer spacing. | [101] |
| | | 2.00 m × 8.00 m<br>3.00 m × 8.00 m<br>4.00 m × 8.00 m<br>5.00 m × 8.00 m<br>6.00 m × 8.00 m<br>6.00 m × 6.00 m | There were significant differences among the treatments in fruit yield but not in fruit quality. The highest (4.15 t ha$^{-1}$) yield was obtained from the distance of 2.00 m × 8.00 m followed by 3.00 m × 8.00 m, 4.00 m × 8.00 m, 5.00 m × 8.00 m, 6.00 m × 6.00 m and 6.00 m × 8.00 m, respectively. | [102] |
| 4 | Guava (*Psidium guajava*) | 2.00 m × 1.00 m<br>3.00 m × 1.50 m<br>3.00 m × 2.00 m<br>4.00 m × 2.00 m<br>4.00 m × 3.00 m<br>4.00 m × 4.00 m<br>5.00 m × 5.00 m<br>6.00 m × 6.00 m | The maximum growth was observed at a planting distance of 6.00 m × 6.00 m. Yield plant$^{-1}$ (17.56 kg) was highest from the distance of 5.00 m × 5.00 m, while fruit yield ha$^{-1}$ was maximum (26.19 t) at 3.00 m × 1.50 m. Similarly, fruit physio-chemical properties (TSS, % TA, and sugar) were significantly reduced with the increasing planting density. | [103] |
| | | 1.00 m × 1.00 m<br>2.00 m × 1.00 m<br>2.00 m × 1.50 m<br>1.50 m × 1.50 m | Plant spacing of 2.00 m × 1.50 m gave significantly maximum (10.25% Brix) TSS, total sugars (10.19%), reducing sugar (5.55%), and non-reducing sugar (4.64%), whereas, lesser plant spacing (1.00 m × 1.00 m) resulted in the significantly lower mean value of above-mentioned quality parameters. A significantly higher yield plant$^{-1}$ (7.28 kg) was obtained in plants spaced at 2.00 m × 1.50 m. but a higher yield ha$^{-1}$ (59.40 t) was obtained with a plant spacing of 1.00 m × 1.00 m. | [104] |
| | | 2.00 m × 2.00 m<br>2.00 m × 1.50 m<br>1.50 m × 1.50 m<br>2.00 m × 1.00 m<br>1.00 m × 1.50 m | The tallest (1.24 m) plant was recorded from the closest (1.00 m × 1.50 m) spacing. The higher number of fruits plant$^{-1}$ (17.20), average fruit weight (77.50 g), yield plant$^{-1}$ (1.32 kg), and TSS/acid ratio (33.14) were recorded under 2.00 m × 2.00 m spacing as against lower value under 1.00 m × 1.50 m spacing. | [105] |
| | | 277 plants ha$^{-1}$<br>312 plants ha$^{-1}$<br>416 plants ha$^{-1}$<br>555 plants ha$^{-1}$ | The average productivity of 6 years was nearly double (15.75 t ha$^{-1}$) in a density of 555 plants ha$^{-1}$ where the planting density was twice as much in recommended spacing (277 plants ha$^{-1}$). | [106] |
| | | 277 plants ha$^{-1}$<br>555 plants ha$^{-1}$<br>1111 plants ha$^{-1}$<br>2222 plants ha$^{-1}$ | Higher density (2222 plants ha$^{-1}$) resulted in the taller plant (5.76 m). An increase in plant density was found to decrease the yield plant$^{-1}$, reduced the fruit weight, and increased the yield ha$^{-1}$. Close planting decreased the TSS and acid ratio of fruit. | [107] |

**Table 2.** *Cont.*

| Sl. | Name of Fruit Crops | Planting Spacing/Density | Key Findings | Reference |
|---|---|---|---|---|
| 5 | Pineapple (*Ananas comosus*) | 35,700 plants ha$^{-1}$ <br> 47,600 plants ha$^{-1}$ <br> 55,500 plants ha$^{-1}$ | Plant height and length of D leaf decreased with the increasing planting density. Fruit length, average fruit weight and total yield ha$^{-1}$ increased with the increasing planting density but quality (TSS and % juice) decreased with the increasing density. | [108] |
| | | 54,000 plants ha$^{-1}$ <br> 66,600 plants ha$^{-1}$ <br> 74,000 plants ha$^{-1}$ | Planting density did not affect the crop growth variables (such as the number of functional leaves and D-leaf length). The planting density did not affect the total weight fruit$^{-1}$, infructescence weight, total fruit length, infructescence length, crown length, or the fruit shelf-life as perceived by traders. The yield increased with an increase in the planting density. | [109] |
| | | 45,000 plants ha$^{-1}$ <br> 55,000 plants ha$^{-1}$ <br> 65,000 plants ha$^{-1}$ <br> 75,000 plants ha$^{-1}$ | The lower planting densities resulted in heavier fruit mass. Yield hectare$^{-1}$ was directly proportional to planting density. Fruit chemical properties were not affected by planting density | [80] |
| | | 64,000 plants ha$^{-1}$ <br> 55,555 plants ha$^{-1}$ <br> 49,382 plants ha$^{-1}$ <br> 45,714 plants ha$^{-1}$ <br> 44,444 plants ha$^{-1}$ <br> 37,037 plants ha$^{-1}$ <br> 29,630 plants ha$^{-1}$ | The highest fruit yield with the crown (52.59 t ha$^{-1}$) was observed in the treatment with the closest spacing (64,000 plants ha$^{-1}$) but the highest fruit weight with the crown (1.29 kg), total sugar (12.56%) and TSS (16.44% Brix) was observed in the treatment with the widest spacing (29,630 plants ha$^{-1}$). | [110] |
| 6 | Banana | 3.00 m × 1.00 m <br> 3.00 m × 1.50 m <br> 3.00 m × 2.00 m <br> 3.00 m × 3.00 m <br> 3.00 m × 3.50 m <br> 3.00 m × 4.00 m | The banana plant at the closest spacing (3.00 m × 1.00 m) had the tallest pseudostem (2.81 m) than under wide spacing. The plants which were spaced at 3.00 m × 3.00 m, gave a higher value of pseudostem girth (84.62 cm) and number of green leaves plant$^{-1}$ (31.3). The highest yield (11.69 t ha$^{-1}$) was produced from plants spaced at 3.00 m × 2.00 m. | [111] |
| | | 1.75 m × 1.75 m <br> 1.50 m × 1.50 m <br> 1.25 m × 1.25 m | The greater value of TSS (23.05%), TSS/acidity ratio (101.90), reducing sugar (14.71%) and total sugars (16.67%) were observed with the closest distance treatment (1.25 m × 1.25 m). | [112] |
| | | 1.20 m × 1.20 m <br> 2.00 m × 2.00 m <br> 2.00 m × 3.00 m <br> 1.80 m × 1.80 m | Growth parameters like plant height (2.06 m), and stem girth (51.98 cm) were found significantly higher with a planting distance of 2.00 m × 2.00 m. Plant growth under a planting distance of 1.20 m × 1.20 m showed a higher yield (117.81 t ha$^{-1}$) as compared to other distances. Higher TSS (21.29% Brix) and total sugar (24.47%) was recorded from 2.00 m × 2.00 m spacing. | [113] |
| 7 | Mango | 5.00 m × 5.00 m <br> 5.00 m × 10.00 m <br> 10.00 m × 10.00 m | The highest plant height (7.00 m), trunk diameter (98.50 cm), number of fruits tree$^{-1}$ (347), fruit weight (271.10 g) and yield tree$^{-1}$ (94.30 kg) were observed in the wider spacing (10.00 m × 10.00 m). The yield ha$^{-1}$ (21.6 t) was higher at closer spacing (5.00 m × 5.00 m). The lowest density (10.00 m × 10.00 m) recorded the highest TSS (19.62%) and acidity (0.18%). | [114] |
| | | 2.50 m × 2.50 m <br> 2.50 m × 5.00 m <br> 5.0 m × 5.0 m | Wider spacing (5.00 m × 5.00 m) gave a significantly higher number of fruits (218 plant$^{-1}$), and fruit yield plant$^{-1}$ (51.78 kg). However, the total yield (34.90 t ha$^{-1}$) was the highest at the closest (2.5 m × 2.5 m) spacing. | [115] |
| 8 | Papaya | 3.00 m × 3.00 m <br> 3.00 m × 2.00 m <br> 3.00 m × 1.00 m | The plant height of papaya did not differ from the plant spacing. However, the highest height of the first flower (87.30 cm), and the height of the fruit set (94.60 cm) of Yuen Nong 1 cultivar were recorded from the closest spacing (3.00 m × 1.00 m). On the other hand, the average fruit weight (2.26 kg) of Maradol cultivar was the highest at the lowest density (3.00 m × 3.00 m). | [116] |
| | | 2.00 m × 1.50 m <br> 2.00 m × 2.00 m <br> 2.50 m × 2.00 m | The highest planting density (2.00 m × 1.50 m) resulted in the tallest (2.12 m) plant. The highest number (72.33 fruits plant$^{-1}$) with the longest (10.93 cm) and widest (8.80 cm) fruit were recorded from the lowest density (2.50 m × 2.00 m). However, the highest number (117,217) of fruits ha$^{-1}$ was recorded from the closest spacing (2.00 m × 1.50 m). | [117] |

## 7. Planting Density on Chlorophyll Content of Fruit Crops

Chlorophyll is essential for plant growth because it facilitates photosynthesis reactions that provide glucose to plants and release oxygen into the atmosphere with the presence of light energy [118]. The chlorophyll content is significantly influenced by planting density. Nagpur mandarin grown under high-density spacing recorded the lowest chlorophyll content, compared to the highest chlorophyll seen at a lower density [99]. Tripathi et al. [119] also observed that the highest chlorophyll content of guava leaf was registered in the widest spacing, whereas the lowest chlorophyll 'a' and 'b' content were found in the closest spacing. The reduction of leaf chlorophyll content at a higher density could be explained partially by the effects of shading of the lower canopy, causing poor canopy interception of the PAR (photosynthetically active radiation). Moreover, the strong interplant competition due to the crowding of the plants may have prevented the absorption of sufficient water and plant nutrients [92].

## 8. Planting Density on Fruit Yield

Crop productivity and optimal resource use are both affected by the spacing between plants [120,121]. Plant density is a key factor in optimizing the structure and increasing the photosynthetic capacity of the plant canopy. Agricultural factors such as crop geometry and plant density influence yield and profitability [122]. Sarrwy et al. [123] reported that the highest yield (13.80 t ha$^{-1}$) of banana was obtained from plants spaced at wider spacing (3 × 2 m) with two suckers per hole, while the lowest yield (12.28 t ha$^{-1}$) was recorded from 3 × 1 m, with one sucker per hole. The earliest flowering (12–13 days) with minimum days required for harvest was also recorded at the widest spacing (3 × 4 m) with three plants per hole, compared to that of the lowest spacing (3 × 1 m), with one

plant per hole. Additionally, the heaviest bunches and fingers with the longest fruit were also harvested from the plants that were spaced at $3 \times 4$ m, with three plants per hole. They mentioned that the highest yield with the wider spacing of three suckers per hole could be attributed mainly to the increased number of plants in a unit area. According to Chaudhuri and Baruah [124], the plant population has a significant impact on both the yield and the yield-related characteristics. The largest bunches (18.50 kg) were found in those with the lowest plant population (4444 plants ha$^{-1}$) and this gradually dropped as the plant density increased. This drop in bunch weight with increasing plant density could be attributed to greater canopy light interception, which may have aided in improving the vegetative characteristics but not bunch weight. In contrast, when the plant population density was low, more leaf surface was exposed to sunlight and, indirectly, a larger amount of assimilates gathered in different organs of the plant owing to the increased bunch weight. Marketable yield was observed to show a tremendous increase per unit area with the increasing plant population density. While the highest plant population (5001 plants ha$^{-1}$: three suckers pit$^{-1}$) produced the highest yield (74.76 t ha$^{-1}$), 3084 plants ha$^{-1}$ with two suckers produced the lowest yield (45.08 t ha$^{-1}$). Plants in lower plant populations had longer, heavier, and thicker fingers than those in higher plant populations. In fig plants, fruit yield in terms of fruit number and fruit weight plant$^{-1}$ increased with the decrease planting density [125]. However, the total yield increased with the increasing planting spacing. Significantly, the highest number, average fruit weight, and yield of mango fruits per tree, including fruit length and breadth, were recorded in spacings of $10 \times 10$ m, but the yield per hectare was higher (216 t) in the spacing $5 \times 5$ m [114]. Planting density had a negative impact on reproductive variables as well. The smaller the area that is available to plants, the greater tendency to decrease the number and yield of fruit per plant. Fruit weight was negatively correlated with plant density and was one of the variables that changed more frequently as planting density increased [126]. As a result, the number of fruits plant$^{-1}$ and their yields decreased.

## 9. Planting Density on Fruit Quality

Fruits are the most important source of our daily nutrition. The quality of fruit is a major aspect from the point of view of consumers. Measures of fruit quality include both internal and external properties. The internal quality is mainly determined by aroma, flavor, taste, texture, and nutritional quality (e.g., soluble sugar content, starch, organic acids, soluble solids content, carotenoids, total flavonoids, total phenolics, antioxidant activity, etc.), flesh firmness, diseases, and chemical residues. The external quality mainly concerns appearance, size, color, and bruises. Fruit quality depends on the distance of planting. Sarrwy et al. [123] confirmed that soluble solids concentration (% Brix) and total sugars (%) were measured from the highest plant spacing ($3 \times 4$ m) with three plants at every hole. Another study by Chaudhury and Baruah [124] reported that the plants that are raised under low density exhibited superior fruit quality in terms of SSC, total sugar, and sugar-to-acid ratio. These parameters tended to decrease as plant density increased. In contrast, titratable acidity and non-reducing sugar showed the opposite trend. The higher acidity in a higher plant population may be due to the shade effect, where sugar conversion from organic acid is hampered due to a lack of sufficient light. Moreover, lower reducing sugar and higher non-reducing sugar in the high-density plot may be due to a lower conversion of sugar from starch. Fruit quality attributes like TSS and acidity of fig were not influenced by tree spacing [125] which was similar to the finding of Mano et al. [127] who reported no difference in fruit quality in trees grown under closer or wider spacing. A better quality of mango in terms of TSS (19.62% Brix), pulp percentage (63.2%), and fruit acidity (0.18%) was harvested from wider spacing ($10 \times 10$ m) when compared to the closer spacing (18.90 and 0.17%, respectively [114]. According to Policarpo et al. [128], under high planting density, besides the changes in the quantity and quality of intercepted light, the partitioning of assimilates between vegetative and reproductive shoots may be responsible for the effects on fruit quality.

## 10. Conclusions

High-density orcharding is one of the recent novel concepts of fruit cultivation, involving the planting of fruit trees densely for better light interception and distribution to increase yield as well as to increase mechanization level. Several studies have been conducted to standardize the planting density of different fruit crops for higher yields with superior quality. A very limited number of studies have focused on an integrated approach of higher planting density with other factors of agronomic management that can improve the quality of fruits. In summary, a lower planting density improved fruit size and fruit quality. However, a higher density of plantings increased yields but reduced the fruit quality. Hence, it is recommended that integrating plant spacing with other cultural practices should be investigated to improve fruit quality.

**Author Contributions:** Conceptualization, S.Z.S. and M.A.H.; literature collection, M.A.H.; writing original draft, M.A.H.; writing—review and editing, S.Z.S. and M.A.H. All authors have read and agreed to the published version of the manuscript.

**Funding:** This research work was conducted with the financial support of the Project Implementation Unit, National Agricultural Technology Programme, Bangladesh Agricultural Research Council, Ministry of Agriculture, People's Republic of Bangladesh, and Universiti Putra Malaysia under vote number 6282524-10201.

**Institutional Review Board Statement:** Not applicable.

**Informed Consent Statement:** Not applicable.

**Data Availability Statement:** Not applicable.

**Conflicts of Interest:** The authors have declared that there is no any conflict of interest.

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
