# Peer review of "Planting Arrangement and Effects of Planting Density on Tropical Fruit Crops—A Review"

_horticulturae, doi:10.3390/horticulturae8060485_

Round 1

Reviewer 1 Report

This is an interesting review on tropical fruit crop densities. I think that the title should include the word "tropical" as this review focusses on tropical fruit crops. If it does not include the word "tropical", it will have to include deciduous fruit crops and various associated trellis systems. Also, in the tropical fruit group, it is missing (especially in Table 2) the major tropical fruit tree crops of banana, mango and papaya (pawpaw) - these should be included in Table 2. Minor tropical crops such as longan, lychee, durian, rambutan, jackfruit, persimmon and mangosteen are also missing but these may not be of global significance.

For an international journal, the units should be converted to metric to be consistent. For example, convert ft to metres and t/acre to t/ha.

In the Introduction, it was mentioned that arable land is the main limitation for fruit production. I would argue that both arable land and water are main limitations for tropical fruit crop production as most fruits contain at least 80% water. The authors also mentioned that low consumption of fruit is due mainly to the reduced cropped area. I would argue that low consumption of fruit is due to both reduced cropped area and poor diets of junk food that are high in calories and low in nutrients.

In addition, I have the following suggestions in the text:

Page 1, Line 8 - Replace "The importance..." with "With the growing importance..."

Page 2, Line 62 - Delete 'for the plant life'.

Page 10, Table 2 - convert units to metric (e.g. ft to metres) to be consistent.

Page 13, Line 398 - convert t/acre to t/ha (convert units to metric to be consistent.

Reviewer 2 Report

The paper falls within the general scope of the journal and gives original review about the subject.
However the paper requires some adjustments/improvements so that I recommend to reconsider it after a minor revision.

Here below some suggestions:

Title: I suggest to change 'potentials' with 'effects'

Line 8-9: check this sentence

Line 160: repetition as Lin 148. Avoid it. My suggestion is to merge the two subparagraphs 2.1.1 and 2.1.2

Information supplied in subparagraphs 2.2.1 and 2.2.2 seem to be overlapped...I am not sure that the two presented pattern of planting are different. Please check and eventually merge the two paragraphs.

Line 193: Add 'and the difficulty of mechanization of some cultural practices.

Line 218: there is no reference to Table 1 in the text.

Subparagraphs in paragraphs 3 and 4 are not necessary. I suggest to remove them. In paragraph 4 please consider also the mechanization level of some practices (pest management, harvest, etc). For some of the considered factors in paragraph 3 and 4 biblio references are not supplied.

Please check Table 2 (reference ion the text?) and all te text for used Units...please check the guideline for authors and change accordingly.

Line 459: add 'and to increase the mechanization level'

Reviewer 3 Report

The objective of the study is interesting when it comes to analyzing fruit trees (evergreen, deciduous, etc.). But the information provided is limited to 5 fruit species, and uses more than 40% of the references of studies carried out on annual crops.

The information provided is broad and not very specific, since it does not provide conclusive information on planting densities that improve the quality of the fruit of specific species.

The conclusions are not responding to the stated objective, it does not identify the research gaps for future considerations.

Reviewer 4 Report

The manuscript entitled " Planting arrangement and potentials of planting density on fruit crops – a review", was elaborated by the authors after reviewing 113 research papers published in the last 20 years. 
This work is in the scope of Horticulturae.
Text included a general introduction, and items regarding the Planting arrangement of fruit crops; The benefit of planting density; factors to be considered for planting density; Impact of plant density on the use of resources; Planting density on the growth and development of fruit crops; Planting density on chlorophyll content of fruit crops; Planting density on fruit yield; and Planting density on fruit quality. Then some concise conclusions.
The manuscript text was well written with simple, clear, and comprehensive technical language, although with abrangent and sufficient detailed information.

Reviewer 5 Report

The review is of obvious interest and relevance. To analyze the problem, the authors used 113 literary sources, of which more than 50% of the work for the past 10 years.

However, the authors did not touch the problem of the influence of rootstock on fruit quality. Most fruit plants are grown with the use of rootstocks. It is recommended to add a section on this topic, which will certainly make the manuscript more integral and interesting for readers.
